# Influence of Pulse–Pause Sequences on the Self-Heating Behavior in Continuous Carbon Fiber-Reinforced Composites under Ultrasonic Cyclic Three-Point Bending Loads

**DOI:** 10.3390/ma15103527

**Published:** 2022-05-13

**Authors:** Aravind Premanand, Frank Balle

**Affiliations:** 1Department for Sustainable Systems Engineering (INATECH), Faculty of Engineering, University of Freiburg, 79110 Freiburg, Germany; frank.balle@inatech.uni-freiburg.de; 2Freiburg Materials Research Center (FMF), 79104 Freiburg, Germany; 3Fraunhofer Institute for High Speed Dynamics, Ernst-Mach-Institute (EMI), 79104 Freiburg, Germany

**Keywords:** self-heating, power-ultrasonics, heat generation, ultrasonic fatigue, carbon fiber reinforced polymers, online-monitoring, 20 kHz, material characterization

## Abstract

Several studies have been conducted in the Very High Cycle Fatigue (VHCF) regime on Carbon Fiber Reinforced Polymers (CFRP) in search of their fatigue limit beyond their typical service life, which is itself in the order of 10^8^ loading cycles. The ultrasonic fatigue test (UFT) method has been recently gaining attention for conducting fatigue experiments up to 10^9^ loading cycles. This can be attributed to the reduction of testing time, as the testing facility operates at a cyclic frequency of 20 kHz. The fatigue loading in UFT is usually performed in a pulse–pause sequence to avoid specimen heating and undesirable thermal effects. For this study, the pulse–pause combination of the UFT methodology was explored and its influence on the self-heating behavior of the CFRP material was analyzed. This was realized by monitoring the temperature evolution in the CFRP specimens at different pulse–pause combinations and correlating it with their final damage morphologies. From the obtained results, it is concluded that the specimen heating phenomenon depends on several variables such as cyclic loading amplitude, the pulse–pause combination, and the damage state of the material. Finally, it is proposed that the test procedure, as well as the testing time, can be further optimized by designing the experiments based on the self-heating characteristic of the composite and the glass transition temperature (Tg) of the polymer matrix.

## 1. Introduction

Approaches such as designing for long life and damage tolerance have become essential with the growing need to make sustainable decisions in regards to engineering applications. One of the standard practices followed by the aircraft industry is to develop and verify the structural design and the manufacturing processes of large composite structures [1]. Successful aircraft programs have used what is called the “building-block approach”, wherein experiments are performed on different materials at coupon level testing and then scaled up to full-scale structural verification. This approach is followed for both metallic and composite structures. However, unlike metals, fatigue of fiber-reinforced composite materials is a more complex phenomenon, owing to their anisotropic behavior [2]. This is because different types of damage (such as fiber fracture, matrix cracking, fiber-matrix debonding, delamination, etc.) can occur autonomously or interactively at different damage growth rates. When fatigue experiments are performed in the laboratory, several parameters can influence the fatigue performance of composites. This includes fiber material and fiber architecture, fiber volume fraction in the laminate, matrix type, stacking sequence of the laminate, environmental condition, and loading condition [3] (which includes the type of loading, stress ratio R, mean stress, and cyclic frequency). Conducting several experiments to take all these variables into account could be expensive and time-consuming. 

Fibers are bundled filaments that can be manufactured with different materials such as glass, graphite, aramid, boron, and silicon carbide, and forms such as short and long fibers. The carbon and Kevlar fibers are usually applied for high-performance components [4,5]. The architecture of fibers can vary from unidimensional forms (unidirectional tows, yarns, and tapes) to bi-(woven and non-woven fabrics) and multi-dimensional forms (fabrics with fibers oriented in more than 2 directions). Fibers also undergo different surface treatments to improve their adhesion to the matrix. The fiber volume fraction depends on the manufacturing method. For example, compression molding results in about 40% of fiber volume fraction, and the filament winding method can result in up to 85% of fiber volume fraction [5]. Polymeric matrix materials can be broadly classified into thermoplastic (polypropylene, polyphenylene sulfone, polyamide, polyetheretherketone, etc.) and thermoset resins (polyesters, phenolics, melamines, silicones, polyurethanes, epoxies, etc.). Different resins are chosen for different applications depending on their properties such as elastic modulus, elongation, coefficient of thermal expansion, type of manufacturing method, and service temperature of the composite material, etc. One of the most important advantages in the design of composites is the ability to control fiber orientation based on the loading that the structure undergoes. Through the stacking of fibers in multiple orientations, efficient load transfer across different directions can be achieved. Depending on the materials used in the composites, several factors such as resistance to corrosion, fluids, creep, and thermal expansion coefficient also plays a role in the fatigue performance of composites. 

Due to these reasons, the fatigue testing programs for different materials are carried out with simple load conditions: in tension or flexure and subsequently with different stress ratios. Later, the advanced fatigue loading conditions such as variable loading or multi-axial loading are carried out for component and demonstrator level testing.

One potential solution to decrease the testing time to reach fatigue lives in the order of 10^8^ cycles during coupon level testing is to increase the cyclic frequency [6]. When increasing the test frequency, investigations need to account for the effect of frequency on the fatigue behavior of the sample material. However, some studies also conclude that frequency would not affect fatigue life if the effect of creep is not significant and the specimen temperature is controlled to stay below the glass transition temperature (Tg) of the polymer [7,8]. 

### 1.1. Effect of Testing Frequency on the Surface Temperature of the Specimen

Fatigue testing methods can be broadly classified into axial and bending tests. Axial tests can be performed in conventional servo-hydraulic machines up to 158 Hz [9] and in ultrasonic test systems at 20 kHz [10,11]. On the other hand, the bending tests can be performed in servo-hydraulic machines up to 158 Hz [9], in cantilever shaker based systems up to 230 Hz [12], in electro-dynamic four-point bending systems up to 80 Hz [13], and in ultrasonic fatigue systems at 20 kHz [14,15,16]. 

Generally, the fatigue experiments in servo-hydraulic systems on polymer composites are carried out at test frequencies in the order of 10 Hz. This is done to avoid the self-heating phenomenon which occurs due to the viscoelastic nature of the polymer matrix in the composite material [17], especially at high stresses. The process of fatigue in composites is accompanied by energy dissipation [18]. Due to the low thermal conductivity of the polymer matrix, the energy is stored in the form of heat. The accumulation of heat with the increasing number of fatigue cycles would cause an increase in the temperature of the specimens. 

Different authors have used different cooling techniques to keep the specimen surface temperature under control. Air cooling and fatigue loading with a pulse–pause sequence are commonly employed for ultrasonic fatigue tests. Balle and Backe (2018) [15] conducted three-point bending fatigue experiments on CF/PPS specimens which resonated at 20.26 kHz with a pulse time of 100 ms and a pause time of 2000 ms to keep the temperature rise below 20 °C. Lee et al. (2019) [19] used a ring-type Peltier cooler with a pulse time of 300 ms and a pause time of 3000 ms to keep the rising temperature below 10 °C for short-glass fiber-reinforced polymer composites under axial loading. Flore et al. (2017) [10] used a pulse time of 100 ms and a pause time of 2000 ms to keep the temperature below 5 °C for glass fiber reinforced composites (GFRP) under axial loading. All these studies employed a constant pulse–pause ratio for all the cyclic amplitudes. 

Even though the operating frequency of the UFT systems is 20 kHz, the effective frequency of the test reduces depending on the pulse–pause sequence. A pulse time of 100 ms and pause time of 2000 ms on a specimen that resonates at 20.26 kHz leads to an effective frequency of 965 Hz. Cui et al. (2020) [16] used a liquid nitrogen cooling system where cold air with a temperature of −39 °C was reported to be blown constantly on the specimen surfaces while maintaining the effective frequency of 19.73 kHz. This means the experiments were carried out without a pulse–pause sequence. In this study, the maximum temperature in the contact regions, which heat up due to friction, was reported to be kept below 40 °C. Even though using a liquid nitrogen cooling system can be useful to substantially increase the testing frequency from 965 Hz to 19.73 kHz, the effect of cooling the specimen surfaces on the fatigue behavior of CFRPs needs further attention. In summary, thermal effects due to accelerated fatigue testing need to be considered to avoid temperature-induced material degradation during fatigue [20].

Lahuerta et al. (2015) [21] studied the combined effect of sample thickness and frequency on the temperature increase of thick GFRP specimens loaded under cyclic compression loads. The thickness was varied between 10, 20, and 30 mm while the cyclic frequency was varied between 0.25 to 2.5 Hz. For the specimens with a thickness of 20 mm and 30 mm, temperature increase beyond 25 °C was observed for higher frequencies. The CFRP specimen thickness for cyclic three-point bending at 20 kHz is ~4 mm [14,16,22], which is smaller in comparison to the 10 mm thick GFRP laminates which did not have a pronounced frequency effect under cyclic compression loads. To reduce the size effect and to make the results of experiments conducted at different frequencies comparable, Lee et al. (2019) [19] conducted axial loading experiments with the same gage volume. For the three-point bending experiments, this would mean that specimens tested at different frequencies should have the same volume between the support units.

### 1.2. Stress Amplitude and Stabilized Self-Heating Phenomenon

Depending on the cyclic load amplitude, the fatigue process may follow one of the following two scenarios concerning the self-heating phenomenon. If the magnitude of the fatigue load is below a certain critical value, the specimen surface temperature increases to a certain value and stabilizes until the start of the final failure phase. This stabilization occurs when the structure can reach a thermal equilibrium such that the generated heat is dissipated to the environment through convection [18]. This phenomenon is called stationary self-heating. In this scenario, the small temperature rise does not influence the fatigue process. Conversely, if the magnitude of the fatigue loads is high enough to generate heat such that the rate of heat generation is greater than the rate of energy dissipation, non-stationary self-heating is observed. In this case, the rise in temperature of the specimen causes mechanical degradation of the material which further increases specimen heating that leads to a significant loss in stiffness and subsequent failure of the specimen. 

This dependency of self-heating on the critical stress amplitude was explored by Rosa et al. (2000) [23] and a methodology was proposed to enable a rapid determination of the fatigue limit for materials. This methodology was termed the Risitano method. Since then, several authors have used this method to determine the fatigue limit for different material systems. Montesano et al. (2013) [24] investigated the fatigue behavior of braided CFRPs by performing conventional fatigue experiments and another methodology termed the “thermographic approach”. The results from the fatigue experiments were used to plot the SN curves. In this study, the Risitano methodology was applied in their thermographic approach where experiments were carried out up to only 7000 fatigue cycles for different cyclic stress amplitudes. The fatigue strength determined by the two approaches was found to be in good correlation. One of the reasons why the Risitano method is successful in the determination of fatigue strengths is related to the stationary and non-stationary self-heating mechanisms described in the previous paragraph. This method assumes that the surface temperature that can be captured through means of infra-red thermography is directly linked to the heat dissipation in the material due to its intrinsic energy dissipation mechanisms. Another approach to verify this argument is to calculate the energy dissipation per unit volume per cycle (hysteresis curve) [25] for different stress amplitudes [24]. The final remarks from these studies are that different cyclic amplitudes lead to different stationary self-heating temperatures depending on the heat dissipation capacity of the material. When the cyclic amplitude is beyond a certain threshold value, non-stationary self-heating occurs. In such a scenario, local temperatures can exceed the Tg of the polymer matrix and eventually alter the condition of damage propagation [26]. This phenomenon is critical for CFRPs due to their viscoelastic behavior, which leads to thermo-mechanical failure. In contrast, if specimens are tested at loading conditions that correspond to stationary self-heating temperatures below the Tg of the polymer material, the temperature rise will therefore not affect the fatigue process in the material.

### 1.3. The Motivation for This Study 

With the possibility of performing accelerated fatigue experiments using UFT systems, the fatigue behavior of different material systems can be analyzed within a relatively short time in comparison to the conventional fatigue experiments which are carried out in servo-hydraulic systems. However, there are several points to be addressed before adopting UFT systems for conducting fatigue experiments in the VHCF regime for composite materials. As noted by Shabani et al. (2020) [6], a clear understanding of the influence of the pulse–pause sequence on the fatigue behavior of composite materials is vital if this methodology is followed for performing fatigue experiments in UFT systems. Further, the temperature rise during fatigue experiments is related to self-heating and damage evolution in the material. However, one important question that needs to be answered is the following: Is it the same at all cyclic amplitudes? As stated by Lahuerta et al. (2015) [21], the temperature that builds up in thick laminates due to volumetric heat generation can be influenced by fatigue frequency, loading ratio, the ratio of strain energy that is transformed into heat, and the boundary conditions of the material that undergoes fatigue loading. 

This study is aimed at capturing the correlation between pulse–pause combination and the underlying fatigue phenomenon in composite materials under cyclic three-point bending. Additionally, a study on the effects of cyclic amplitude on the self-heating behavior of fiber-reinforced polymers was deemed necessary to avoid unnecessary thermal effects. Thus, a strong understanding of the limiting boundaries of the self-heating phenomenon would enable the further development and optimization of ultrasonic fatigue testing methodology for damage characterization in CFRPs.

## 2. Materials and Methods

### 2.1. Composite Material

Thermosets and thermoplastics are two major categories of a polymer matrix for composite materials. Thermoplastics fail at higher strains in comparison to thermosets [27]. Some of the other major advantages of thermoplastics for the composite material are unlimited shelf life, low consolidation duration, and recyclability, which makes them suitable for high-performance applications that also require durability. In recent years, the applicability of CF/PEKK composite material to large structures such as aircraft fuselage shells [28] and the torsion box for horizontal stabilizers [29,30], and small structures such as ribs [31] (which are elemental to aircraft wings) were successfully demonstrated by the Thermoplastic Affordable Primary Aircraft Structure (TAPAS) Consortium. Thus, the carbon fiber reinforced in Polyether–ketone–ketone (CF/PEKK) composite material manufactured by Toray Advanced Composites (Nijverdal, The Netherlands) was chosen for this work. PEKK is a semi-crystalline thermoplastic polymer that belongs to the family of Polyaryl–ether–ketone (PAEK), with a melting point temperature (T_m_) of about 340 °C and a glass transition temperature (Tg) of about 160 °C. This makes it suitable for elevated-service temperature applications, such as in the aerospace industry, where high mechanical performance is also necessary. The material was manufactured by the consolidation of a 5-harness satin fabric and PEKK polymer matrix with a fiber volume fraction of 50%, which resulted in an overall mass density of 1.53 g/cm^3^. 

The absence of manufacturing defects such as voids and delaminations was confirmed via the ultrasonic C-scans provided by the manufacturer. The stacking sequence of the layup is [0_f_/90_f_/0_f_/90_f_/0_f_/90_f_/0_f_]_s_. The subscript “f” denotes the orientation of the warp fiber bundles of the fabric and “s” denotes symmetry. The rightmost 0_f_ layer in the center is not included in the symmetry and so is denoted with an underscore. Thus, 13 layers of prepregs resulted in an overall consolidated laminate thickness of 4.1 mm. This can be visualized in Figure 1c, which depicts the optical micrograph of the thickness face of the laminate.

### 2.2. Specimen Geometry

To conduct ultrasonic fatigue loading on CFRP specimens, suitable geometry needs to be designed such that the specimen oscillates in its first transversal bending eigenmode due to resonance at 20 kHz [14]. This was determined using Ansys workbench 17.1 software with a homogeneous orthotropic material model based on six elastic moduli and three Poisson’s ratios (see Table 1) and the mass density. Some of these elastic constants were determined using standard monotonic tests according to DIN EN ISO 527-4 and DIN EN ISO 14129. Other values were estimated using the micromechanics model developed at the University of Twente [32]. The modal analysis from the finite element software resulted in a specimen geometry of 34 × 15 × 4.1 mm^3^ (see Figure 1b). The span between the support units, which can be seen in Figure 1a, was 18.6 mm. The distance between the nodal displacement points which can be visualized in Figure 1a, where relative displacement is zero, was chosen as the span length between the support units to reduce frictional heating during resonance oscillations.

### 2.3. Monotonic Three-Point Bending Experiments 

Monotonic bending experiments were performed with the Zwick/Roell Z010 test system (Ulm, Germany) with a load cell of 10 kN on specimens designed for the VHCF experiments to determine the monotonic failure load and the corresponding mode of failure. For the determination of flexural properties of fiber-reinforced polymers under three-point bending loads, a minimum span-to-thickness ratio of 16:1 is recommended by the ASTM standard D7264/D7264M-15. This is not possible for the specimens designed for the VHCF experiments, as the geometry of the specimens is realized from the modal analysis which resulted in a span-to-thickness ratio of 4.5:1. Hence, it is not applicable to directly use the maximum flexural stress and strain equations proposed in the standard. This is because shear deformation occurs for specimens tested with span-to-thickness ratios less than the recommended standard [33]. 

The applied force and the crosshead displacement were recorded by the machine. All the experiments were carried out under displacement control, with a displacement rate of 1 mm/min. Additionally, a combination of Canon EOS 80D camera and a lens with a focal length of 60 mm was used for capturing the deformation during monotonic loading. Further, with the use of the Digital Imaging Correlation (DIC) technique, it was possible to realize the strain distribution in the samples due to three-point bending loads just before failure. A speckle pattern with an average speckle diameter of ~5 pixels was applied using an airbrush. The image data for DIC analysis were processed using the NCorr library inside the MATLAB environment. 

### 2.4. Fatigue Three-Point Bending Experiments

An in-house UFT system was developed using the ultrasonic load train manufactured by Herrmann Ultraschalltechnik GmbH (Karlsbad, Germany). A schematic of the loading system and the online monitoring units, along with their model names, are provided in Figure 2a. The loading system consists of a 4.8 kW digital ultrasonic generator, a piezoelectric convertor, an amplitude booster unit, and a load introduction unit (loading nose). This combination leads to sinusoidal oscillations with a maximum amplitude of up to ±57 μm when 100% power is applied by the ultrasonic generator. This displacement is then transferred to the VHCF specimen which undergoes bending (see Figure 2b) due to the two supports at the bottom face of the specimen. Since the specimen is not directly connected to the load train, a monotonic deflection greater than the cyclic oscillation is required to ensure permanent contact between the specimen and the loading nose during fatigue loading [14]. One of the current challenges in the VHCF testing of composites in UFT systems is the inability to capture the real-time stresses and stress distribution during cyclic oscillations at 20 kHz. This is because a suitable force sensor requires a sampling rate of greater than 40 kHz to capture the peak-to-peak force amplitudes. Indirect measurement of stresses is, however, possible through non-contact measurement of strains and if the stiffness of the material is known. For this study, the cyclic displacement amplitude and mean displacement amplitude are used to define the fatigue cycle.

During the fatigue experiments, the cyclic oscillation of the specimen was monitored using a single-point laser Doppler vibrometry (LDV) of type CLV2534 from Polytec GmbH (Waldbronn, Germany) at a frame rate of 500 kHz. The changes in the surface temperature of the specimen due to ultrasonic oscillations were monitored using an infra-red (IR) camera TIM640 from Micro-Epsilon Messtechnik GmbH (Ortenburg, Germany). Thermal images of the temperature distribution in the fatigue specimens were captured at a resolution of 640 × 480 pixels. A control unit was employed to handle the ultrasonic oscillation parameters from the generator inside the LABVIEW environment. 

A monotonic displacement of 200 μm was applied for all the experiments. This monotonic displacement was measured through means of a digital dial-gage indicator (MarCator 1086R) from Mahr GmbH (Göttingen, Germany) that captures the deflection at the bottom center of the specimen. 

Constant amplitude tests were performed at three different cyclic amplitudes (40 μm, 45 μm, and 50 μm) and three different pause times (1 s, 2 s and 4 s), which resulted in a total of 9 experiments (see Figure 3a). Figure 3 provides the characteristic pulse–pause sequence with the data obtained from the LDV. To reach the desired cyclic amplitudes without an overshoot, an onset (ramp-up) time of 20 ms was chosen (see Figure 3b). The part of the pulse for actual fatigue cycles was set to 200 ms (see Figure 3c), and finally, an off-set (ramp-down) time was set to 30 ms. Depending on the resonance frequency of the specimens, which was ~20,207 Hz, cyclic oscillation for 200 ms resulted in ~4040 fatigue cycles per pulse.

The fatigue experiments were set to terminate either when the maximum surface temperature at the contact point between the specimen and the support units exceeded 80 °C (which is 50% of Tg of PEKK polymer) or when the resonance frequency of the ultrasonic system which excites the specimen dropped below 19.5 kHz or when the total fatigue life of the specimen reached 10^8^ cycles. Since the compressed air is blown on the surface of the specimens which can be assumed as thick beams, it was assumed that a surface temperature of 80 °C would mean a higher internal temperature. Further, as the scope of the research was to determine the effect of the pulse–pause sequence on damage initiation and propagation, it was deemed to be sufficient to test the specimens until the fatigue life reached 10^8^ cycles.

### 2.5. Preparation of Samples for Microscopy

After the end of the monotonic and fatigue experiments, the specimens were cut into two halves and embedded inside an epoxy polymer, as shown in Figure 4. This was done to observe both the edges of the specimens. The cured epoxy samples which contain the cut VHCF samples were then ground and polished for microscopy using the Struers LaboPol-30 system (Willich, Germany). Figure 4 shows an overview of the observed specimen surfaces. The damage morphologies were observed using a digital light optical microscope (Model name: Smartzoom 5) from ZEISS (Oberkochen, Germany).

## 3. Results and Discussion

### 3.1. Specimen Response under Monotonic Loading

The visualization of a typical strain distribution, produced through DIC analysis just before the final failure of the VHCF specimens, can be seen in Figure 5. The maximum normal strains were found in the top (compressive strain) and bottom layers (tensile strain) of the specimens (see Figure 5a), while the maximum shear strains were found in the region between the loading nose and the support units (see Figure 5b). 

These specimens failed at a loading force of 7330 ± 63 N, corresponding to a central deflection of 1103 ± 17 μm. The final failure was the same for all the specimens and this was found to be due to the facture of the fibers in the bottom face of the specimen due to tension (see Figure 6a). This can also be seen in Figure 6b where the damage initiation at the bottom layer was followed by transverse damage propagation to the top side of the specimen. This also implies that the monotonic bending stresses are more critical than the monotonic shear stresses in the CF/PEKK material for the chosen specimen geometry.

Since bending failure was observed, the apparent maximum bending strength was calculated using the homogeneous beam theory, as suggested by ASTM D7264/D7264M-15:(1)$$\sigma =\frac{3PL}{2bh^{2}}$$
where *P* is the maximum monotonic load, *L* is the span length between the supports which is 18.6 mm, *b* denotes the width, and *h* is the thickness of the specimen. The monotonic bending strength was found to be 811.05 ± 6.97 MPa.

### 3.2. Specimen Response under Fatigue Loading

Table 2 provides a summary of the number of cycles to failure or run-outs for the specimens tested at different combinations of cyclic amplitudes and pause times. 

Looking at just the number of cycles to failure, one can conclude that higher cyclic amplitude leads to failure. For the case of the specimens tested at 50 μm, the lower pause times led to a shorter fatigue life. However, this trend was not the same for specimens tested at 45 μm where the specimen tested at a pause time of 2 s had a failure earlier than the specimen tested at a pause time of 1 s. One should also acknowledge that failures also scatter in HCF and VHCF regimes. Lastly, for the case of specimens tested at the cyclic displacement of 40 μm and at different pause times, no failure was observed until 10^8^ cycles. In the following subsections, the characteristics of these experiments are analyzed in detail.

#### 3.2.1. Influence of Pulse–Pause Combination on the Self-Heating Behavior of VHCF Specimens

In the VHCF experiments, it was observed that the regions of the specimens near the support units underwent self-heating due to friction at the contact point of the specimens and the support units. Figure 7 provides a typical temperature distribution along the length of the specimen in the top, middle, and bottom lines drawn on the surface of the undamaged specimen as the effect of an ultrasonic pulse.

From Figure 7, it can be interpreted that the bottom regions where the specimen is in contact with the two support units are more critical than in the other regions. It is interesting to note that heating does not occur at the top line where there is contact between the loading nose and the specimen. This is because the loading nose and specimen are oscillating at resonance while the bottom supports are stationary. This causes relative motion between the specimen and the support units at the points of contact, which results in frictional heating.

Furthermore, the temperature profile along the bottom line is seen to be asymmetric. This depends on the sensitivity of the material against friction at the locations of contact with the support units. Since the area on the specimen close to the contact points can be interpreted to be critical due to the self-heating phenomenon, the maximum temperature in these regions was monitored for all the specimens. 

Figure 8 shows the time-temperature plot captured using IR thermography at a frame rate of 20 Hz for the three different cyclic amplitudes and different pause times. Here, the change in maximum surface temperature near the contact points against time is plotted. A window of one minute was chosen, as it enables the visualization of temperature stabilization for all the specimens, as well as the pulsed oscillation behavior of the temperature curves in the form of a saw-tooth. 

One could argue that the frame rate of the IR camera is much less in comparison to the frequency of ultrasonic pulses and it does not allow for capturing accurate temperature information within the pulses. However, in the case of pulse–pause loading, the temperature drop during the pauses can be considered an important parameter. This is because the heat generated during the ultrasonic pulses is trapped if the pause times are too short to cool down the surface of the specimen. In other words, even though the infra-red camera does not have the sufficient frame rate to precisely capture the ultrasonic pulses, which last for 200 ms, this is adequate to capture the cooling behavior during the pause times which last for 1, 2, and 4 s. 

The peaks in each of these curves can be related to the ultrasonic pulses and the valleys can correspond to the ultrasonic pauses. In the case of the specimen tested at a cyclic amplitude of 40 μm and 4 s pause time, the longer pause time allows the maximum temperature to cool down such that the heat generated during a pulse is dissipated before the start of the next pulse. This resulted in a temperature rise of less than 5 °C. In the case of the specimen tested at a cyclic amplitude of 50 μm and a pause time of 1 s, which is placed on the top of the graph where the pause time is too short, the heat energy generated during a pulse was not dissipated and the sequential ultrasonic pulses led to higher maximum surface temperature. 

It is also worthwhile to note that the stabilization temperature for the specimen tested at a cyclic amplitude of 45 μm and a pause time of 1 s is close to the stabilization temperature of the specimen tested at a cyclic amplitude of 50 μm and a pause time of 2 s, where the temperature rise is about 18 °C. From this observation, it can be interpreted that the self-heating temperature is dependent on both the cyclic amplitude, as well as the pause time between the pulses. This can be understood in detail in Figure 9. In Figure 9, the change in peak temperature within the pulses on the specimen surface near the contact regions is plotted as a function of fatigue cycles for all the VHCF specimens.

For the specimens tested at different cyclic amplitudes with a pause time of 4 s, the overall temperature rise was about 7 °C. This means a pause time of 4 s allowed the specimens to cool down with the temperature rise only as a result of the cyclic amplitude. However, for the case of specimens tested at a pause time of 2 s, a cyclic amplitude of 40 μm resulted in a temperature rise of about 10 °C, while a cyclic amplitude of 50 μm doubled the temperature rise to about 22 °C. 

One hypothesis that could be made from this observation is that the strain energy is applied to the specimen during the loading part of the fatigue cycle and the quantity of energy that is applied during a fatigue cycle is related to the amplitude of cyclic displacement. In other words, more energy is applied for a higher amplitude and vice-versa. Now, the energy that remains at the end of an ultrasonic pulse needs to be dissipated in the form of heat or through damage growth. Thus, the observation of different stabilization temperatures for the case of the specimens tested at 2 s pause time can be attributed to the difference in the amount of strain energy supplied to the specimens at different cyclic amplitudes. This also explains the case of the specimen tested at 50 μm at a pause time of 4 s; even though higher strain energy is supplied to the material during loading, the pause time between successive pulses is sufficient enough for the material to dissipate the remaining energy that was not returned during unloading. 

For the specimen tested at a cyclic amplitude of 50 μm and a pause time of 1 s, a gradual temperature rise to 26 °C was observed. After this, an onset of damage near the bottom face of the specimen coupled with the non-stationary self-heating effect (see Figure 10d) led to a sudden rise in the temperature to about 35 °C, followed by the final failure of the specimen. 

The characteristic features that were captured using IR-thermography during the fatigue experiments are shown in Figure 10. 

Figure 10a presents the first scenario where the increase in surface temperature, especially at the contact points, was below 20 °C throughout the entire test. This was observed for specimens tested with pause times of 2 and 4 s at all cyclic amplitudes until the onset of damage, and for specimens that had a run-out. In the second scenario, the surface temperature at the location of damage increased during the onset of damage. This can be visualized in Figure 10b where heating occurred due to delamination in the specimen where the bending stresses were high. The temperature in the other regions, including those near the contact points, did not significantly heat up, indicating that the effect of temperature rise due to the self-heating phenomenon and the damage can be differentiated. This behavior was observed in the specimens tested at a cyclic amplitude of 50 μm at the pause time of 4 s and for the specimens tested at a cyclic amplitude of 45 μm and pause times of 1 and 2 s. 

The phenomenon of non-stationary self-heating was observed for the specimen tested at a cyclic amplitude of 50 μm and pause time of 1 s is shown in Figure 10c. In this case, the region close to the contact points was observed to heat up with the increasing number of cycles without temperature stabilization. Furthermore, the temperature of the entire specimen rose in the case of a non-stationary self-heating phenomenon, which is different in comparison to Figure 10a,b. Finally, the scenario depicted in Figure 10d was observed for the same specimen depicted in 10c just before its failure at 6.67 × 10^7^ cycles. Here, the non-stationary self-heating phenomenon—due to short pause time as well as heating due to damage—was observed. Additionally, the maximum increase in surface temperature at the contact region was about 35 °C (see Figure 9) and the temperature increase in the region of damage was observed to exceed 80 °C, after which the experiment was immediately terminated. 

From the experimental VHCF data, it can be concluded that self-heating behavior for CF-PEKK specimens is critical for higher cyclic amplitudes with low pause times. Further, during the onset of damage, the loss in stiffness of the specimen can lead to an increase in the self-heating temperatures, which can then interact with the damage, leading to a further increase in the temperature of the specimen. 

Going forward with the results of this study, online monitoring of surface temperature evolution at the bottom middle region of the specimens, along with the temperature evolution monitoring near the contact regions, is proposed. Additionally, a detailed analysis of the effect of heat dissipation at different distances from the damage or failure sites using structural evaluation methods such as spectroscopy and X-ray computed tomography [18,34] is recommended and planned for the future by the authors.

#### 3.2.2. Damage Morphologies Observed during Stationary and Non-Stationary Self-Heating Fatigue

The specimens that failed within 10^8^ cycles, as well as the specimen that had a run-out at a cyclic amplitude of 40 μm at a pause time of 1 s, were prepared for microscopy, as described in Figure 4. For the specimen that had a run-out, only the microscopic features, such as longitudinal and transverse cracks, were identified in the regions between the supports and the loading nose; this can be seen in Figure 11. 

All specimens that failed within 10^8^ cycles were observed to have a fracture of fiber bundles and delamination in the top (in the region of high compressive strains) and bottom layer (in the region of high tensile strains) of the specimens. Moreover, the macroscopic features were observed to be similar on both sides of the specimens. Figure 12 provides the delamination observed in the specimen tested at a cyclic amplitude of 50 μm at a pause time of 2 s. 

For the delamination that occurred at the top and bottom layers, the epoxy that was used for embedding the specimens can be seen to have flown inside the broken areas, thus enabling the delaminated area to become dark gray in color, which is different from the PEKK polymer that is light gray in color.

For the specimen tested at 50 μm and a pause time of 1 s, apart from the delamination in the top and bottom layers, a cohesion failure of the PEKK polymer was observed in the location corresponding to high surface temperatures in Figure 10d. Here, the fiber fracture and the non-stationary self-heating phenomenon could have interacted to locally soften the polymer material which can be seen in Figure 13b. The black-colored features in Figure 13b correspond to cohesive matrix cracks. In the top layer of the specimen, delamination near the upper face of the specimen caused the flow of epoxy during the embedding process; this can be seen in Figure 13a. 

Several interpretations could be made from the micrographs described in Figure 11, Figure 12 and Figure 13. Specimens with the run-outs still have microscopic damage features which do not propagate within 10^8^ cycles. For all the failed specimens, damages were predominantly concentrated in the top and bottom layers. Unlike the monotonic failure (Figure 6b) where the damage propagated transversely across the specimen from the bottom to the top, delamination was observed along the longitudinal direction for the fatigue specimens. This caused a loss of stiffness for the specimens, and hence their loss of ability resonated at 20 kHz. Only in the case where the specimen was tested at 50 μm and a pause time of 1 s was the friction due to damage suspected to interact with the non-stationary self-heating behavior, resulting in cohesive matrix cracks near the location of 0° fiber fracture. This is the same location where the temperature exceeded 80 °C before the interruption of the experiment, and such a feature was only observed near the failure location. One can interpret that a damage feature such as matrix cracking due to cohesive failure could have resulted from the combined effect of mechanical fatigue and elevated temperature. However, to validate or eliminate such a claim, further investigations are needed on the fatigue of CF/PEKK specimens where self-heating temperatures reach 0.5 × Tg. It is also suggested that in the presence of thermo-mechanical degradation behavior, such tests are invalid for the determination of fatigue strength. 

Interestingly, this was not observed for all the other cases, including the specimen tested at 50 μm and a pause time of 2 s, even though the overall temperature rise (Figure 9) was greater than 20 °C. This means that for a cyclic amplitude of 50 μm, a pause time ≥ 2 s would provide valid results.

Katunin and Wronkowicz (2017) [35] also observed that the non-stationary self-heating phenomenon occurs in glass–fiber reinforced polymers under cyclic bending after the maximum increase in temperature exceeds a certain threshold of self-heating temperature (60 °C for this study). It was also observed that the self-heating phenomenon occurred after the temperature rise beyond 60 °C became non-linear and the final failure was dominated by thermal effects. For the specimens that failed with maximum self-heating temperature values below 60 °C, the damage characteristics were purely related to mechanical degradation due to the fatigue phenomenon. 

The next step after the experiments was to perform damage propagation analysis under ultrasonic fatigue loading. This requires in situ microscopy where the polished surface of the specimen under monotonic loading could be captured during the pause times after the onset of damage. These micrographs can be used to identify the local strain fields in the vicinity of damage and be matched with the corresponding thermal data to provide further insights into the correlation between heating and damage.

## 4. Conclusions and Outlook

The heating rate of a specimen during sequential pulse–pause loading is dependent on the combined influence of the cyclic amplitude and pause time. Higher cyclic amplitudes result in shorter fatigue lives, and it is the lower cyclic amplitudes that require a long testing time. However, what is the right combination of pulse–pause sequence to carry out fatigue experiments in the UFT systems for composite materials? The answer is that there is no generic solution that can be applied to all polymer composites. With a pause time of 1 s, the test time to reach 10^9^ cycles is less than 4 days. The same test takes 12 days when tested with a pause time of 4 s. This shows why pause time is critical for conducting VHCF experiments in ultrasonic fatigue test systems. 

If the VHCF experiments are conducted for an analysis of damage initiation mechanisms, one can say that the combinations of cyclic amplitudes and pause times that lead to a temperature rise of within 20 °C (in the contact regions with the support units) do not contribute to thermal effects. This implies that lower cyclic amplitudes can be tested up to higher fatigue lives by decreasing the pause times. For example, for cyclic amplitudes less than 40 μm, a pause time of 1 s can be chosen as the overall temperature rise of less than 20 °C. However, the validity of such a statement for other materials is based on several conditions:The fiber and fiber architecture should enable the dissipation of heat such that it is not trapped in the bulk of the material.The glass transition temperature of the polymer is high enough such that specimen softening does not occur when the surface temperature of the specimen during ultrasonic fatigue loading rises to 20 °C.Fatigue damage mechanisms do not interact with self-heating mechanisms until the onset of damage.

In the presence of macroscopic damage such as delamination, frictional heating occurs at the location of delamination. There is a possibility that the localized heating can interact with the self-heating phenomenon, causing a significant rise in the temperature of the specimens. Thus, it is interpreted that the controlling of pause times based on the self-heating temperature is valid until the onset of damage.

Further analyses of microscopic damage features are proposed for the specimens loaded with monotonic displacement instead of embedding them inside an epoxy polymer. This is possible if the specimens are polished before conducting the fatigue experiment and with the use of in situ microscopy during UFT.

The current study was performed by controlling the cyclic amplitude during ultrasonic fatigue loading. In contrast, fatigue cycles are commonly defined in terms of stresses—namely, mean stress and stress amplitude. Direct measurement of stresses during ultrasonic oscillations cannot be adequately measured for CFRP specimens with the current state-of-art force sensors. To measure at least two data points per cycle, the force sensor requires a frame rate greater than 40 kHz. Even then, two data points per cycle are insufficient to define a fatigue cycle which is a sinusoidal wave. One possible solution to determine the stresses corresponding to the cyclic displacement amplitudes is to measure the strain distribution during cyclic loading at 20 kHz. This can be achieved with the use of instruments such as a 3D scanning laser vibrometer and a high-speed camera, as these devices can operate at a frame rate greater than 100 kHz. Once the strains are measured, the corresponding stresses can be determined by using Hooke’s law for orthotropic materials. The bending stress versus fatigue life diagram (also known as a Woehler diagram) can then be plotted, for the studies that follow. 

## Figures and Tables

**Figure 1 materials-15-03527-f001:**
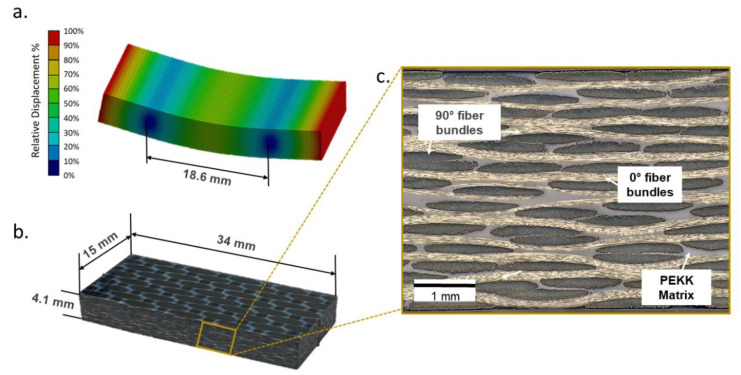
(**a**) Transverse bending oscillation mode at 20 kHz determined using Ansys workbench software with the distance between nodal displacement points equal to 18.6 mm; (**b**) dimensions of CF-PEKK specimen for realizing resonance at 20 kHz; (**c**) a micrograph of the undamaged specimen before cyclic loading to visualize the stacking sequence.

**Figure 2 materials-15-03527-f002:**
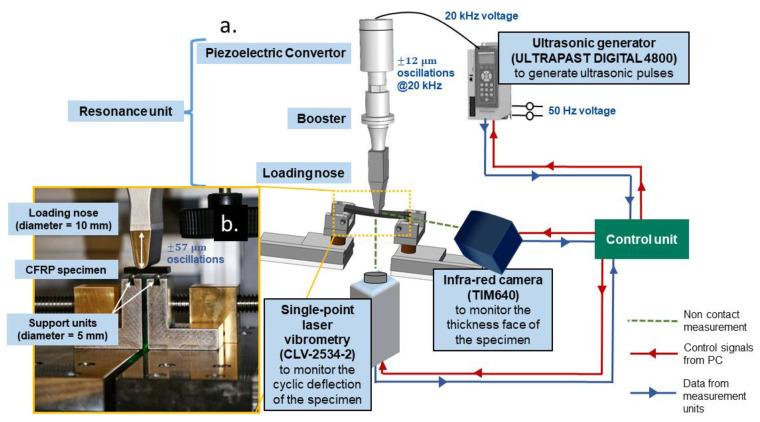
(**a**) A schematic of the UFT system and the online monitoring devices at INATECH; (**b**) a closer view of the experimental setup for applying three-point bending loads.

**Figure 3 materials-15-03527-f003:**
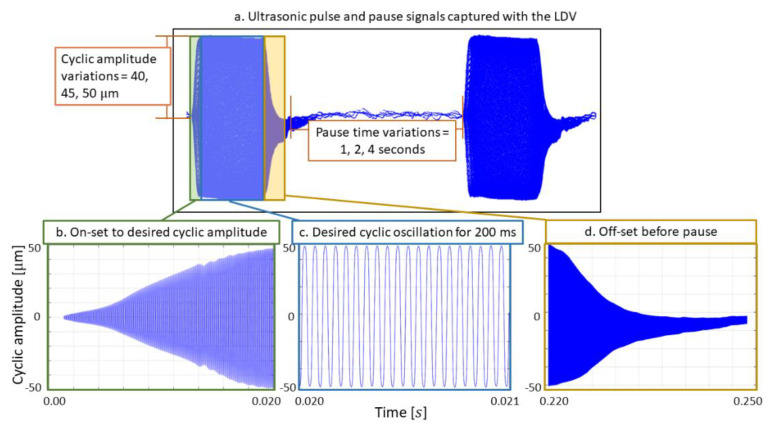
(**a**) A characteristic ultrasonic pulse−pause sequence captured by the LDV; (**b**) the on−set regime (ramp−up) of the pulse; (**c**) the part of the pulse with cyclic amplitude = 50 μm; (**d**) off−set (ramp−down) regime of the pulse.

**Figure 4 materials-15-03527-f004:**
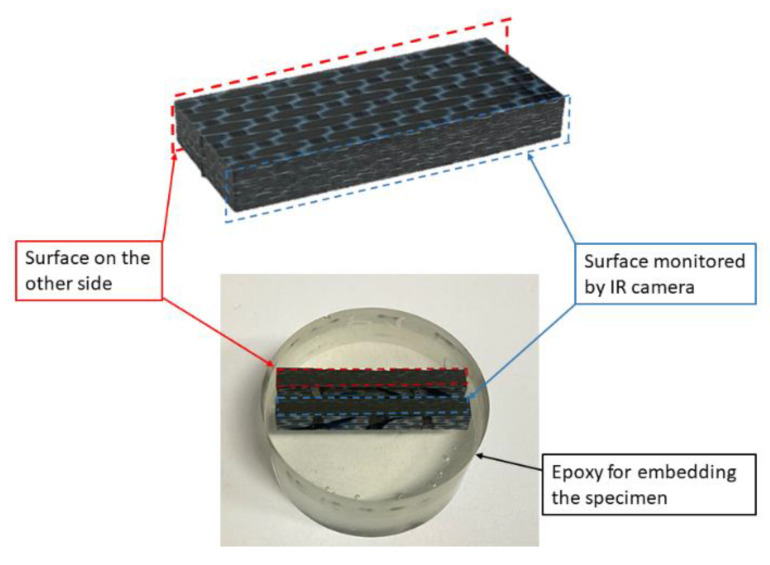
An overview of a typical specimen embedded inside polymer for microscopy.

**Figure 5 materials-15-03527-f005:**
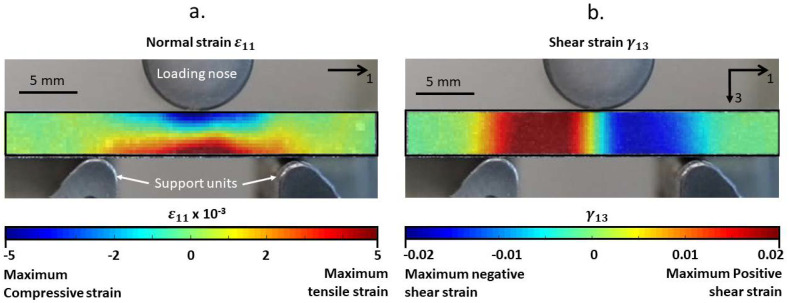
The strain distribution in the VHCF specimen under monotonic three−point bending load just before failure was calculated using DIC: (**a**) normal strain and (**b**) shear strain.

**Figure 6 materials-15-03527-f006:**
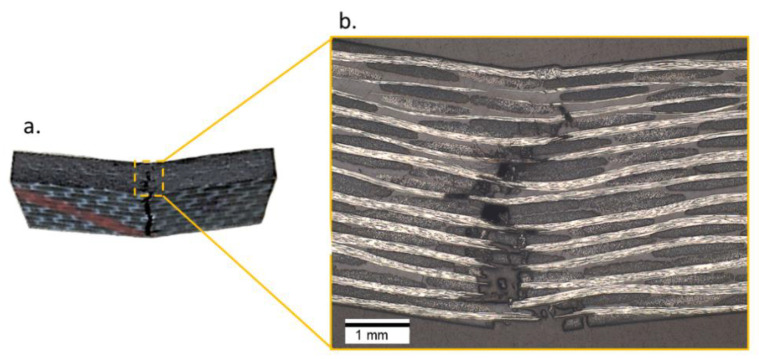
(**a**) Monotonic failure due to high tensile strain in the bottom layers of the specimen and (**b**) a closer look at the crack propagation location captured by the light optical digital microscope.

**Figure 7 materials-15-03527-f007:**
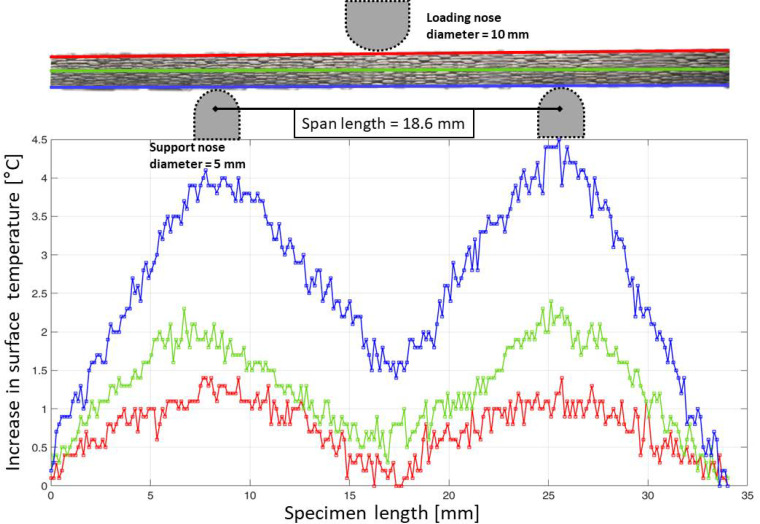
Characteristic temperature profile across the length of the undamaged specimen in the top, middle, and bottom regions of the specimen surface when tested at a cyclic amplitude of 40 μm and pause time of 4 s.

**Figure 8 materials-15-03527-f008:**
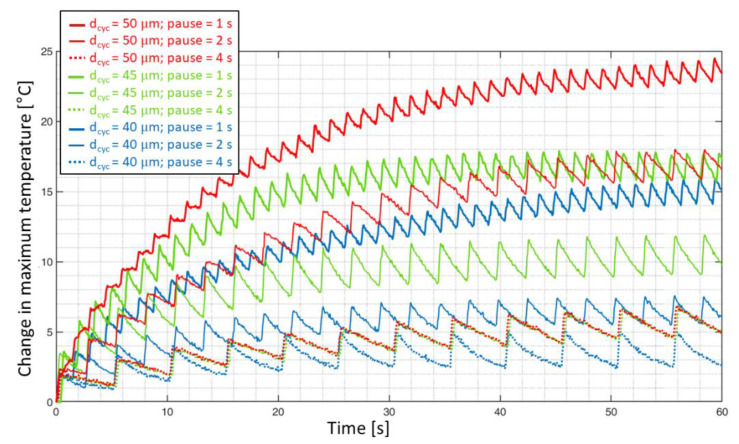
A time–temperature plot of the VHCF specimens that were tested at 20 kHz in three different load amplitudes and three pause times during the first minute.

**Figure 9 materials-15-03527-f009:**
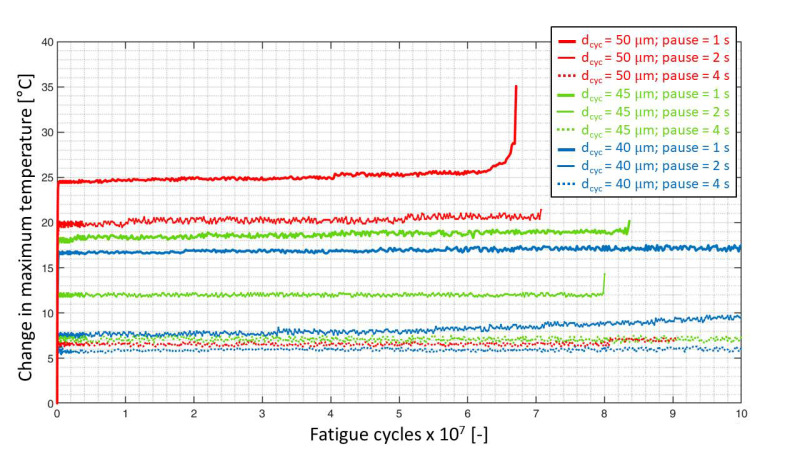
The change in maximum surface temperature near the contact regions against the fatigue life of the VHCF specimens tested at 20 kHz in three different load amplitudes and pause times.

**Figure 10 materials-15-03527-f010:**
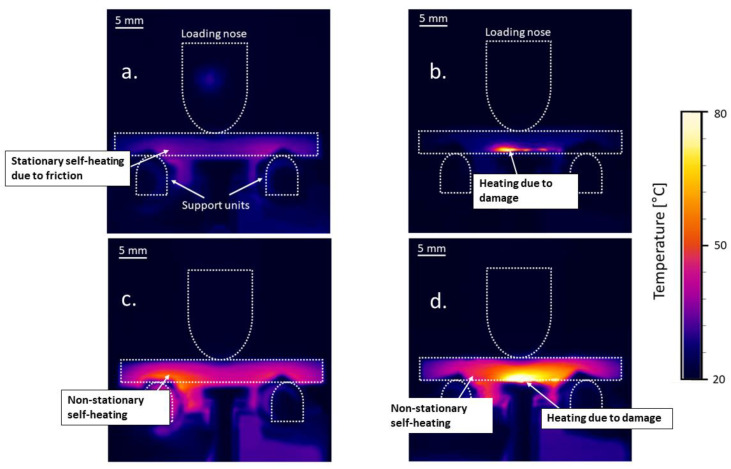
The characteristic features observed during the ultrasonic fatigue loading of VHCF specimens tested at different cyclic amplitudes and pause times: (**a**) stationary self−heating due to friction between the specimen and support units; (**b**) heating in the presence of damage; (**c**) non−stationary self-heating; (**d**) combination of non-stationary self−heating and damage.

**Figure 11 materials-15-03527-f011:**
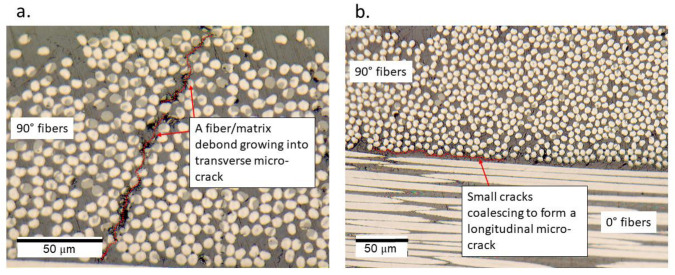
(**a**) Transverse microcrack, and (**b**) longitudinal microcracks in the specimen that was tested at a cyclic amplitude of 40 μm and a pause time of 1 s.

**Figure 12 materials-15-03527-f012:**
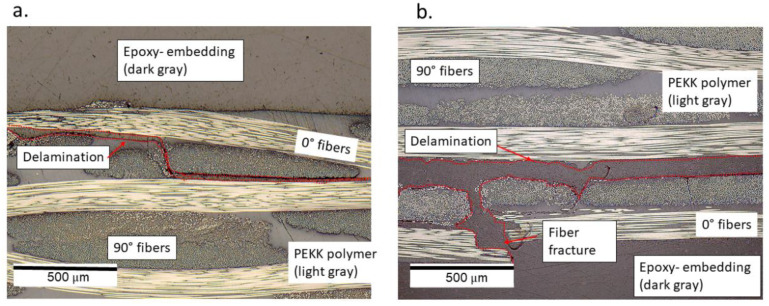
The delamination observed (**a**) in the top layer and (**b**) in the bottom layer of the specimen tested at a cyclic amplitude of 50 μm and a pause time of 2 s.

**Figure 13 materials-15-03527-f013:**
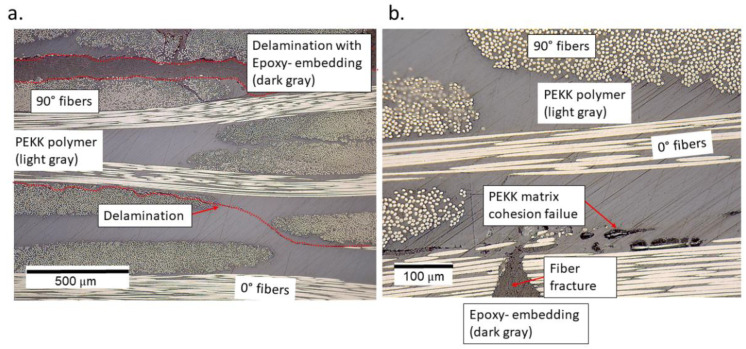
The delamination observed (**a**) in the top layer and (**b**) in the bottom layer of the specimen tested at a cyclic amplitude of 50 μm and a pause time of 1 s.

**Table 1 materials-15-03527-t001:** The elastic constants for the orthotropic model used in ANSYS software for the modal analysis of CF/PEKK material.

Elastic Properties	Values
E11 [GPa]	58.00
E22 [GPa]	58.00
E33 [GPa]	13.00 *
G12 [GPa]	4.80
G13 [GPa]	5.70 *
G23 [GPa]	5.70 *
ϑ12 [-]	0.05
ϑ13 [-]	0.5 *
ϑ23 [-]	0.5 *

* Values marked with * are estimated using the micromechanics model developed at the University of Twente [32].

**Table 2 materials-15-03527-t002:** The number of cycles to failure and run-outs observed for specimens tested at different cyclic amplitudes and pause times at 20 kHz.

Cyclic Amplitude [μm]	Pause Times [s]	Failure [Cycles]/Run-Out *
50	4	9.07 × 10^7^
50	2	7.07 × 10^7^
50	1	6.71 × 10^7^
45	4	Run-out
45	2	8.00 × 10^7^
45	1	8.38 × 10^7^
40	4	Run-out
40	2	Run-out
40	1	Run-out

* Run-outs in the context of this analysis meant the absence of delamination until 10^8^ cycles.

## Data Availability

The data presented in this study are available on request from the corresponding author.

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
