# Peer review of "Influence of Pulse–Pause Sequences on the Self-Heating Behavior in Continuous Carbon Fiber-Reinforced Composites under Ultrasonic Cyclic Three-Point Bending Loads"

_materials, 2022, doi:10.3390/ma15103527_

Round 1

Reviewer 1 Report

This paper investigated the influence of pulse-pause sequences on the self-heating behavior in continuous carbon fiber-reinforced composites under ultrasonic cyclic three-point bending loads. The ultrasonic fatigue test (UFT) method was adopted to accelerate the fatigue tests. The influence of pulse-pause combination on the self-heating behavior was analyzed. The damage morphologies during fatigue were observed. Overall, the paper is well organized and presented. The following comments can further improve the quality of the paper.

* Introduction

#1 The reviewer suggests authors should provide the information on the type, properties, composition, and the sensitivity to the cyclic loading of fiber-reinforced composite materials in the first paragraph in the introduction. Please see the recent work: type/properties: Composite Structures, 2022. 281: 115060. sensitivity to the cyclic loading: Polymer Composites, 2018, 39(6): 1785-1808.

2# The increase of surface temperature of composites during the fatigue is not only related to the test frequency, but also related to the size of the sample. For general large-size specimens, the temperature rises too much due to the untimely heat dissipation during fatigue loading. It is suggested that the authors increase the relevant summary on the temperature increase related to the sample size and testing frequency. For example, International Journal of Fatigue, 2019, 120:141-149.

3# When summarizing the effects of testing frequency and stress amplitude on the surface temperature and self-heating phenomenon, how do the authors consider the formation of heat and the increase of temperature in the anchoring system at both ends of FRP. Compared with the sample surface, the anchorage system belongs to a closed state. Heat cannot be dissipated in time, which may lead to a significant increase in temperature, the softening of resin and anchoring failure of FRP. It is suggested that the authors add relevant summary and analysis of heat generation in the anchorage system in the above two parts.

4# What is the gap between the accelerated fatigue experiments using UFT systems and actual stress condition during the application? Can the acceleration method in this paper truly simulate the stress state of FRP in the actual service?

*Materials and Methods

#1 For the samples for microscopy, it is the control or after the monotonic/fatigue three-point bending experiments? The latter is more meaningful to reveal the fatigue failure mechanism. In addition, the sample preparation process for microscopy should be briefly described.

*Results

1# In Figure 5, DIC is used to monitor the strain distribution during three-point bending loading. The current nephogram can not clearly reflect the strain. It is suggested to convert the nephogram into strain distribution curve for the key areas, which is more meaningful.

2# Please provide flexural strength for the three-point bending loading until failure.

3# What is the effect mechanism of cyclic amplitude and pause time on fatigue damage? Please add relevant fatigue failure mechanism analysis. Why is not DIC used during the fatigue? This is important to reveal the fatigue damage mechanism of composites.

4# During the fatigue process, it can be found that there are multi-damage in the material. Can the authors relate these damages to the relevant parameters of fatigue test?

5# How to evaluate the bending fatigue life of composite?

Author Response

Dear Editor / Reviewer,

Please find our response to the comments of reviewer 1 in the attachment.

Kind regards,

Aravind Premanand

Reviewer 2 Report

In this study, the authors presented original and interesting studies on very high cycle fatigue of polymer matrix composites. The main impact of this study was focused on thermal effects occurring during specific loading sequence proposed by the authors. The topic is very actual in terms of fatigue testing of composites. In the Introduction, the authors presented a broad state-of-the-art review on fatigue of composites and accompanying thermal effects, pointing on a problem of excessive heating of composite specimens during testing. Based on this, they presented a motivation of their research study. In section 2, the authors described a tested material as well as experimental setup for very high cycle fatigue testing and further characterization of tested specimens. Further, in section 3, the results of the study were presented and discussed. A deep analysis of thermal behavior and fracture mechanisms of tested specimens lead to a formulation interesting and important hypotheses for proper understanding thermomechanical behavior of composites under loading frequencies at ultrasonic range and perform a general evaluation of influence of particular parameters of testing on validity of the proposed accelerated testing approach. The study is valuable and draws a background for further investigations planned by the authors.

1) Please provide information on specific applications of a tested material and its exposition to fatigue loading to confirm a legitimacy of selection of this material for tests.

2) Please comment on the assumed criterion on exceeding of 50% Tg temperature as an ultimate one. With the thickness of 4.1 mm and a presence of an air-blowing system, it is expected that the internal temperature of a composite is much higher than the surface temperature.

3) How the assumed cyclic amplitudes correlate with accompanying stress values?

4) The caption of Figure 9 seems to be incorrect.

Author Response

Dear Editor / Reviewer,

Please find our response to the comments from reviewer 2 in the attachment.

Kind regards,

Aravind Premanand

Round 2

Reviewer 1 Report

Although the authors provided a cover letter, more comments have not been responded, and some necessary information should be added in the revised version. The specific comments as follows.

* Introduction

Point 1: Without the basic introduction of fiber-reinforced composite materials, it is very abrupt to directly mention its fatigue properties. The above basic information is very helpful for readers to understand the performance of FRP and its engineering application background. Therefore, the reviewer suggests that the authors consider the first review to provide the basic information about the type (CFRP, GFRP and BFRP etc.), properties (lightweight, excellent mechanical properties, corrosion/creep/fatigue resistances), composition (fiber, polymer and interface) of FRP. 

Point 2: As known, the three-point bending fatigue testing has the limited specimen dimension. However, the summary of the introduction not only covers the current main research work, but also some other factors should be further summarized and analyzed. For example, the effects of sample size and testing frequency on the increase of temperature increase are very important during the fatigue process. Therefore, these factors should be further considered in the introduction.

Point 3: The authors responded “In ultrasonic fatigue test systems, the cyclic oscillation occurs due to resonance between the loading unit and the specimen. This is different in comparison to the accelerated fatigue tests that are performed in the servo-hydraulic machines where the system and the specimen operate without resonance.”. However, whether the ultrasonic fatigue test system can be simulated the working conditions faced in the real service environment? Furthermore, the heat in the anchor can generate during the ultrasonic fatigue testing.

*Results

Point 1: In Fig. 5, there are only two labels marked as blue and red. Therefore, we can’t get the strain quantitatively for the other colors. For example, what is the strain value of light green?

Point 4: For the first question: “During the fatigue process, it can be found that there are multi-damages in the material. Can the authors relate these damages to the relevant parameters of fatigue test?”, the reviewers did not see a point-to-point response.

Point 5: Why is there no S-N curve in the current paper? This is very important for evaluating the fatigue life of composites.
